# Preparation and Characterization of Chitosan–Nano-ZnO Composite Films for Preservation of Cherry Tomatoes

**DOI:** 10.3390/foods10123135

**Published:** 2021-12-17

**Authors:** Yu Li, Yu Zhou, Zhouli Wang, Rui Cai, Tianli Yue, Lu Cui

**Affiliations:** College of Food Science and Engineering, Northwest A&F University, Xianyang 712100, China; liyu_915@163.com (Y.L.); zhouyuy@tju.edu.cn (Y.Z.); wzl1014@nwsuaf.edu.cn (Z.W.); cairui@nwsuaf.edu.cn (R.C.); yuetl@nwafu.edu.cn (T.Y.)

**Keywords:** chitosan, nano-ZnO composite films, preservation, cherry tomatoes, antibacterial activity

## Abstract

Chitosan is widely used as a natural preservative of fruits and vegetables, but its poor mechanical and water resistances have limited its application. Therefore, in this study, we prepared chitosan composite films by incorporating different amounts of nano-zinc oxide (nano-ZnO) to improve the mechanical properties of chitosan. We also assessed the antibacterial activity of these films against selected microorganisms. The addition of nano-ZnO improved the tensile strength (TS) and elongation at break (EAB) of the chitosan films and reduced their light transmittance. TS and EAB increased from 44.64 ± 1.49 MPa and 5.09 ± 0.38% for pure chitosan film to 46.79 ± 1.65 MPa and 12.26 ± 0.41% for a 0.6% nano-ZnO composite film, respectively. The ultraviolet light transmittance of composite films containing 0.2%, 0.4%, and 0.6% nano-ZnO at 600 nm decreased from 88.2% to 86.0%, 82.7%, and 81.8%, respectively. A disc diffusion test showed that the composite film containing 0.6% nano-ZnO had the strongest antibacterial activity against *Alicyclobacillus acidoterrestris*, *Staphylococcus aureus*, *Escherichia coli*, and *Salmonella*. In a 15-day preservation study, chitosan composite films containing 0.6% nano-ZnO maintained the soluble solid content of cherry tomatoes, effectively inhibited their respiration, and exhibited good antibacterial properties against the selected microorganisms. Overall, the prepared chitosan nano-ZnO composite film showed a good preservation effect on cherry tomatoes.

## 1. Introduction

Cherry tomato (*Solanum lycopersicum*) is an edible vegetable grown worldwide; it is a rich source of vitamins A and C, and the main source of lycopene and *β*-carotenoids in the human diet [1]. According to the statistical database of the Food and Agriculture Organization, China’s annual tomato production reached 59,514,773 tons in 2017, accounting for nearly 33% of the world’s total production; thus, China is by far the largest cherry tomato producer, followed by USA and India [2]. As tomatoes have thin skins and a high liquid content, they can easily lose water and shrink during storage and transportation; their freshness decreases rapidly after harvest, resulting in great economic losses [3]. Cherry tomatoes are often eaten fresh, such that pathogens on the surface can cause food-borne diseases. Therefore, it is important to improve cherry tomato preservation measures during transportation and storage.

Packaging is the key to ensuring food safety and quality, reducing the effects of spoilage organisms, as well as chemical and physical hazards, and thereby slowing food deterioration and prolonging storage time [4]. Recently, environmentally friendly packaging materials composed of biopolymers have been increasingly developed based on polysaccharides, lipids, and proteins. Chitosan is the deacetylation of chitin. As an important natural polysaccharide, chitosan has been widely used in biomaterials, healthcare, environment and other industrial fields due to its good biocompatibility, antibacterial activity, and safety [5]. At present, chitosan and its modified materials have been widely applied in food preservation and packaging. For example, chitosan-coated (3% and 4%) smoked herring samples displayed more than 4 log10 CFU/g reduction in aerobic plate count, along with the complete suppression of yeast, mold and *Enterobacteriaceae* count [6]. Meanwhile, chitosan can maintain the phenolic and flavonoid compounds along with the physiochemical quality of smoked fish during frozen storage. A novel procyanidins-loaded chitosan-graft-polyvinyl alcohol film was prepared and applied to sustain the antibacterial activity for food packaging [7]. The obtained film exhibited a desirable biofilm inhibition and antibacterial activity against spoilage bacteria and foodborne pathogenic microbes. For salmon preservation, the film can prevent microorganism contamination and texture deterioration in 10 days. However, their poor mechanical properties and water resistance limit their applications in food packaging.

Biopolymer production can involve the addition of active compounds, such as polysaccharide and nanoparticles [8,9]. These nanoparticles improve the structure and mechanical properties of biocomposite films. Among inorganic nanoparticles, zinc oxide (ZnO) nanoparticles are the most promising nano-agents due to their remarkable physical and chemical properties. The addition of nano-ZnO improves the antibacterial and antioxidant activities of the packaging system, extending the shelf life of foods packaged using these products [10]. Nano-ZnO is nontoxic to human cells and harmful to microorganisms. The U.S. Food and Drug Administration (FDA) has listed nano-ZnO as a safe material [11]. Siddiqi et al. reported that nano-ZnO with 100 μg/mL was toxic, but nano-ZnO with low concentration was safe [12]. Additionally, Jayasuriya et al. reported that the composite films containing 1% nano-ZnO (30 nm) did not show any cytotoxicity of cells; the concentration of nano-ZnO over 5% exhibited apparent toxicity [13]. However, decreasing the nano-ZnO particle size can improve the antibacterial activity of prepared materials [14]. Several studies have shown that chitosan–nano-ZnO composite films are effective in maintaining fruit color, reducing water loss, increasing antibacterial activity, and extending the freshness period [4]. For example, P. M. Rahman et al. demonstrated that a chitosan–nano ZnO composite film significantly extends the shelf life of meat [15].

In this study, a series of chitosan–nano-ZnO composite films were prepared and their mechanical, optical, water solubility and antibacterial properties were evaluated. Then, the optimal composite film was selected, and its characterizations were further measured. Finally, we examined the effects of chitosan–nano-ZnO composite films on cherry tomato preservation during postharvest storage, as well as their antibacterial effects on Gram-negative and Gram-positive bacteria. Through this experiment, we will provide new materials for fruit and vegetable preservation.

## 2. Materials and Methods

### 2.1. Materials

Cherry tomatoes were purchased from a local market. Chitosan powder was purchased from Solarbio Science & Technology Co, Ltd. (Beijing, China). Nano-ZnO powders (50 ± 10 nm, 99.8% purity) were obtained from Aladdin Biochemical Technology Co, Ltd. (Shanghai, China). Tween 80 and glycerol were obtained from Chemical Reagent Factory (Xi’an, China). *Alicyclobacillus acidoterrestris* (*A. acidoterrestris*, DSM 3923) was purchased from the German Resource Centre for Biological Material (DSMZ, Germany). *Staphylococcus aureus* (*S. aureus*, ATCC25923), *Escherichia coli* (*E. coli*, ATCC25922), and *Salmonella* (ATCC13076) were purchased from the American Type Culture Collection (USA). Brain Heart Infusion (BHI) broth was obtained from Land Bridge Technology Co, Ltd. (Beijing, China). Lysogeny broth (LB) culture was prepared using 10 g tryptone, 5 g yeast extract, and 10 g NaCl. *A. acidoterrestris* medium (AAM)was prepared using 2 g yeast extract powder, 2 g glucose, 0.4 g (NH_4_)_2_SO_4_, 1.0 g MgSO_4_·7H_2_O, 1.2 g KH_2_PO_4_, 0.38 g CaCl_2_, and 1 L distilled water. *E. coli* and *Salmonella* were incubated in LB medium, and *S. aureus* was incubated in BHI medium; both were cultured at 37 °C for 6–8 h with shaking at 120 rpm. *A. acidoterrestris* was incubated in AAM and cultured at 45 °C for 24 h with shaking at 120 rpm. The obtained bacterial solution, with a concentration of about 10^7^ CFU/mL, was used in the inhibition test.

### 2.2. Preparation of Composite Films

All films were prepared using the solution casting technique. To prepare a 1% chitosan solution, chitosan was added to a 2% (*v*/*v*) glacial acetic acid solution, which was then stirred at 30 °C until the chitosan had completely dissolved. Glycerin (0.25%) and Tween 80 (0.1%) were added to the solution; after stirring, different amounts of nano-ZnO (0.2%, 0.4%, 0.6% (*w*/*v*)) were added, followed by ultrasonic defoaming for 30 min. The resulting composite film solutions had various concentrations (0, 0.2%, 0.4%, and 0.6%) of nano-ZnO. Finally, 20 mL of each prepared solution was added to a 10 cm × 10 cm culture dish and air dried at room temperature to obtain uniform films. The characterization and antimicrobial properties of the prepared films were further evaluated.

### 2.3. Properties of Chitosan Containing Different Nano-ZnO

In order to obtain suitable materials that can be used for preservation, the prepared chitosan composite films containing different amounts of nano-ZnO were measured.

#### 2.3.1. Mechanical Properties

The flat, smooth composite films were cut into 2.5 cm × 8 cm strips, and their tensile strength (TS) and elongation at break (EAB) were measured using a texture analyzer (TA.XT-PLUS, Stable Micro Systems, London, UK). The original spacing and stretching rates were set at 5 cm and 60 mm/min, respectively. Five replicates per sample were prepared and the average was calculated. TS was calculated as the ratio of the maximum tensile force to the cross-sectional area, as follows:(1)$$\mathrm{TS}=F_{\max}/\left(L\times W\right),$$
where TS is the tensile strength (MPa), F_max_ is the maximum tensile force (N), L is the film thickness (mm), and W is the film width (mm).

EAB (%) was calculated as follows:(2)$$\mathrm{EAB}\%=\left(L_{1}-L_{0}\right)/L_{0}\times 100\%,$$
where L_0_ is the initial film length (mm) and L_1_ is the length of the film when it breaks (mm).

#### 2.3.2. Optical Properties

The optical properties of the chitosan–nano-ZnO composite films were evaluated according to the light transmittance of the films. Briefly, the prepared films were cut into 4 cm × 1 cm rectangles using a utility knife and placed in a quartz cell. An empty quartz cell was used as a blank control. Film transmittance was measured at a wavelength of 600 nm using an ultraviolet (UV) spectrophotometer (UV-1700, Shimadzu, Kyoto, Japan) with three replicates.

#### 2.3.3. Water Solubility

Films pieces (20 × 20 mm) were dried to constant weight in an oven at 105 °C for 24 h. Subsequently, the films were immersed in 50 mL of water at room temperature (20 ± 5 °C). After 24 h of immersion, the samples were treated in an oven at 105 °C for 24 h until reaching constant weight. The solubility was calculated as follows:(3)$$\mathrm{Solubility}\left(\%\right)=\frac{m_{1}-m_{2}}{m_{1}}\times 100,$$
where m_1_ is the initial dry weight (g) and m_2_ is the final dry weight (g).

#### 2.3.4. Antimicrobial Properties

The antimicrobial properties of composite and chitosan films were determined using the agar diffusion method [16]. The Gram-negative bacteria *Salmonella* and *E. coli*, and the Gram-positive bacteria *S. aureus* and *A. acidoterrestris* were selected for evaluation. The film samples (round disks; diameter = 6 mm) were placed on a clean bench, for UV light sterilization for 30 min prior to bacteriostatic measurement. Bacterial culture solution (200 μL) was spread on the solid plate medium, and the test film was placed in the center of the plate. The inhibition zone was measured to determine the antimicrobial activity of the composite films after the bacteria had been incubated for 48 h. Each sample was measured in triplicate, and the average was calculated.

### 2.4. Characterization of Composite Film Materials

Based on the mechanical, optical, water solubility and antibacterial properties of different composite films, chitosan composite film containing 0.6% nano-ZnO was selected and its characterizations were further measured.

#### 2.4.1. X-Ray Diffraction (XRD)

The XRD patterns of composite films were obtained using XRD (Ultimate IV, Rigaku, Tokyo, Japan) with Cu Ka radiation (λ = 1.5418A) at 40 kV and 40 mA. The samples were analyzed between 10 and 80° with increments of 2°/min. The composite films with a thickness of approximately 20 μm were prepared for the XRD test.

#### 2.4.2. Scanning Electron Microscopy (SEM)

The microstructure and morphology of the composite films were observed using SEM (S-3400N, Hitachi, Tokyo, Japan). Each film was cut into small pieces and fixed on a sample table using conductive double-sided tape. Gold sputtering was performed using an MSP-IS ion sputtering instrument. The samples were observed at an accelerating voltage of 5 kV.

### 2.5. Composite Film Application Study

#### 2.5.1. Evaluation of Quality Parameters

In our study, cherry tomatoes with good appearance, no mechanical damage and with no visible mold were selected. Each group containing nine cherry tomatoes was used for the evaluation of quality parameters. Cherry tomatoes were rinsed in tap water for 5 min and dried at room temperature. Then, the cherry tomato samples of the coating group were immersed in chitosan solution with 0.6% nano-ZnO, shaken gently for 1 min, and air-dried in a biological safety cabinet for 1 h. The control groups were immersed in distilled water, and the other steps remained the same as the coating group. Each group of cherry tomatoes were then stored at 20 ± 5 °C and 70–75% relative humidity for 15 days. The physicochemical properties of cherry tomatoes were evaluated every three days during storage. The respiratory rates (mg CO_2_/kg sample/h) of the cherry tomato samples were measured in a closed drying oven using a fruit and vegetable respiration rate meter [17]. The color for the surface of cherry tomatoes was measured using a colorimeter (Ci7600, X-Rite, Grand Rapids, Michigan, USA) and evaluated in terms of lightness (*L**), red/green content (*a**), and yellow/blue content (*b**). The total color difference (ΔE) was calculated as reported previously [18]. The soluble solid content (SSC) of cherry tomato samples was measured by a saccharometer corrected using distilled water at 20 °C. Cherry tomatoes were crushed in a blender, homogenized and then measured by saccharometer. Briefly, a homogenate sample solution was dropped on a prism aimed at a light source, and SSC was directly measured [4]. The ascorbic acid content of cherry tomato samples was measured using a commercial kit provided by Jiancheng Bioengineering Institute (Nanjing, China).

#### 2.5.2. Inoculation Antimicrobial Assay

Cherry tomatoes were rinsed in tap water for 5 min, immersed in sterile distilled water for 10 min, and then wiped with sterile cotton balls containing 75% ethanol. Each group contained nine cherry tomatoes. The cherry tomatoes were immersed in different inoculation solutions, shaken gently, and placed in a biological safety cabinet and dried at room temperature for 1 h. Then, each sample was immersed in chitosan solution, shaken gently for 1 min, and air-dried in a biological safety cabinet for 1 h. This experiment included a control group without film packaging. All cherry tomatoes were stored at room temperature (20 ± 5 °C) for 15 days and measurements were taken every 3 days. The samples were placed in 100 mL 0.1% (*w*/*v*) sterile buffered peptone water (BPW) for 2 min homogenization and then serially diluted. The diluted bacterial suspension was spread on plates and cultured for 48 h, and the bacterial concentration of each strain was then detected using the conventional spread-plate method.

### 2.6. Statistical Analyses

All statistical analysis was performed using SPSS 18.0 (IBM, Armonk, NY, USA). The obtained data were reported as the mean ± standard deviation (*n* ≥ 3). We performed one-way analysis of variance (ANOVA) followed with Duncan’s multiple test to evaluate significant differences between treatments (*p* < 0.05). All plots were created using OriginPro 8.0 software (OriginLab, Northampton, MA, USA).

## 3. Results and Discussion

### 3.1. Choice of Chitosan Film with Optimal Nano-ZnO Concentration

#### 3.1.1. Mechanical Properties

In our study, the particle size of nano-Zno was 50 ± 10 nm and the maximum content was 0.6%, the prepared chitosan–nano-ZnO composite film should be safe. The mechanical properties of composite films can improve the quality and stability of perishable food products during transport and storage. As shown in Figure 1, the TS and EAB values of pure chitosan film were 44.64 ± 1.49 MPa and 5.09 ± 0.38%, respectively, whereas those of composite films containing different concentrations of nano-ZnO were in the ranges 32.78 ± 2.23–46.79 ± 1.65 MPa and 6.66 ± 0.66–12.26 ± 0.41%, respectively. Compared with the pure chitosan film, the TS values of composite films containing 0.2% and 0.4% nano-ZnO decreased by 26.57% and 1.21%, while that of the composite films containing 0.6% nano-ZnO increased by 4.82%, respectively. The TS values of composite films containing 0.4% and 0.6% nano-ZnO were not significantly different (*p* > 0.05). The EAB values of composite films containing 0.6% nano-ZnO were significantly higher (*p* < 0.05) than those of the pure chitosan film and composite films containing 0.2% and 0.4% nano-ZnO. The EAB values of composite films containing 0.2%, 0.4%, and 0.6% nano-ZnO increased by 30.84%, 40.39%, and 140.86%, respectively. Thus, the addition of nano-ZnO enhanced the TS of the films, consistent with previous report. Sanuja et al. concluded that the TS and EAB values of chitosan films increased by adding 0.1–0.5% ZnO [19]. These findings can be explained by the formation of strong new bonds between chitosan and ZnO due to nanoparticle–matrix interface interactions [20]. The addition of nano-Zno can generate an intermolecular cross-linking effect. The interface bonding between chitosan and nano-particles results in the effective transfer of stress to the nano-particles, which can improve the mechanical properties of the composite film [21].

#### 3.1.2. Optical Properties

The optical properties of functionalized composite film can affect the sensory quality of packaged fruits and vegetables. Transmittance measurements of the composite films showed that the transparency of pure chitosan film was 88.2%, whereas those of composite films containing 0.2%, 0.4%, and 0.6% nano-ZnO were slightly reduced, at 86.0%, 82.7%, and 81.8%, respectively. There were obvious differences between each group of composite films and pure chitosan film (*p* < 0.05). This optical change was caused by high UV absorption and scattering by ZnO [22], which subsequently generated electron hole pairs. Thus, the composite film offered significant UV protection and good transparency. Further studies should explore its potential effects on the sensory quality of fruits and vegetables.

#### 3.1.3. Water Solubility

The water solubility values of the composite film containing 0, 0.2%, 0.4%, and 0.6% nano-ZnO were 25.4 ± 1.25%, 22.15 ± 1.2%, 19.75 ± 2.2% and 18.45 ± 0.98%, respectively. The results indicated that the water solubility of chitosan film gradually decreased with the addition of nano-ZnO. This may be caused by the cross-linking of nano-ZnO and the hydrophilic groups on the chitosan chain, thereby weakening the interaction between the hydrophilic groups and surrounding molecules [15].

#### 3.1.4. Antimicrobial Properties

Chitosan has broad-spectrum antibacterial properties and high film-forming stability, which can be degraded by microorganisms. The addition of nano-ZnO to the chitosan composite film significantly enhanced its antibacterial effects. The inhibition zone diameters and disc diffusion study in different strains was shown in Figure 2A,B, where a, b, c and d represent the strains of *Salmonella*, *E. coli*, *A. acidoterrestris* and *S. aureus*, respectively. The inhibition zones of pure chitosan film against *Salmonella*, *E. coli*, *S. aureus*, and *A. acidoterrestris* measured 9.85 ± 0.50, 9.37 ± 0.36, 9.30 ± 1.13, and 6.13 ± 0.20 mm, respectively, perhaps due to the antimicrobial properties of chitosan. The disc diffusion study showed that all films had significant antibacterial effects on the selected microbial strains (Figure 2B). Chitosan forms porous structures on the surfaces of Gram-positive bacterial cells, causing the chitosan to bind to the cell membrane and destroy its barrier function. Chitosan also penetrates Gram-negative bacteria to adsorb its ionic substances and interfere with metabolism, thus inhibiting bacterial growth. Compared with pure chitosan film, the inhibition zones of the four tested bacteria increased as the amount of nano-ZnO increased, in a manner dependent on the nano-ZnO concentration. The antimicrobial activity of composite films containing 0.6% nano-ZnO showed appreciable difference (*p* < 0.05) towards four strains when compared to the pure chitosan film (Figure 2A). The addition of 0.6% nano-ZnO significantly increased the inhibition zones of *Salmonella*, *E. coli*, *A. acidoterrestris*, and *S. aureus* to 23.30 ± 1.06, 16.36 ± 1.26, 52.25 ± 3.35, and 11.62 ± 0.54 mm, respectively, and the antibacterial effect of the composite film was significantly higher against *A. acidoterrestris* than against *Salmonella* and *E. coli*. The sensitivity of Gram-positive bacteria to nano-ZnO is higher than that of Gram-negative bacteria, whereas the antibacterial effect of the composite films against *S. aureus* observed in this study was in contrast to that reported in a previous study [23]. This may be caused by the presence of a thick layer of peptide glycans in the cell wall of *S. aureus* [24]. Nano-ZnO is a good antibacterial agent and appears to greatly improve the antibacterial properties of chitosan film. ZnO enhances the positive charges of the chitosan amino group, which enhances interactions with the negatively charged microbial cell wall [15], thus potentially exerting a synergistic effect in nano-ZnO composite films.

The effects of various concentrations of nano-ZnO (0, 0.2%, 0.4%, and 0.6%) on their mechanical, optical, water solubility and antibacterial properties are evaluated in the above. The TS and EAB values of 0.6% nano-Zno composite films were increased by 4.82% and 140.86%, respectively. Compared with pure chitosan film, the mechanical properties were effectively improved. The optical properties of each composite films were basically above 80%, indicating that the compatibility between chitosan and nano-ZnO was good. The water solubility of chitosan film gradually decreased with the addition of nano-ZnO. The antibacterial assay further showed that the increase of nano-ZnO content can improve the antimicrobial properties of composite films. Based on the best performance of mechanical properties and antibacterial properties, the composite film containing 0.6% nano-ZnO was used for the subsequent experiments.

### 3.2. Characterization of Composite Film Materials

#### 3.2.1. XRD Analysis

The XRD patterns of pure chitosan and composite (0.6% nano-ZnO) films are shown in Figure 3. The typical peaks of chitosan (A and B) appeared at 19.9°, which showed its semicrystalline nature. Figure 3C showed the XRD pattern of nano-ZnO powder. The characteristic peaks of nano-ZnO appeared at 31.6°, 34.3°, 36.2°, 47.5, 56.5°, 62.8°, and 67.8°, which corresponded to (1 0 0), (0 0 2), (1 0 1), (1 0 2), (1 1 0), (1 0 3), and (2 0 1) planes of nano-ZnO, respectively [25]. However, the XRD peaks corresponding to the presence of 0.6% nano-ZnO were difficult to observed in Figure 3D, which may be due to the low concentration of nano-ZnO in the coating film. Zhang et al. concluded that a higher concentration of the nano-ZnO in composite film meant the more characteristic peak of nano-ZnO [26], while the diffraction peaks were not observed nanocomposite films with low concentrations of nano-ZnO [27]. In addition, the chitosan peak shifted and appeared at 12.7°, which presumably due to the heterogeneous effect of chitosan. Zhang et al. reported that the chitosan peak of CTS/nano-ZnO composite film with contents of 0.2%, and 0.3% all shifted [26].

#### 3.2.2. SEM Images

SEM images of the pure chitosan and composite (0.6% nano-ZnO) films showed different surface morphologies. The pure chitosan film had a uniform surface and texture (Figure 4A), whereas the smoothness of the composite film surface disappeared because of nano-ZnO accumulating and dispersed over the surface of chitosan (Figure 4B). These results suggested that the two polymer materials were compatible, and the film structure was changed by the addition of nano-ZnO. Qiu et al. reported that there were uniform ZnO protrusions in the composite films due to the fixation effect from the interaction between chitosan and Zn^2+^ [28]. Additionally, the addition of nano-ZnO has been reported to interfere with the molecular structure of chitosan, and to affect its mechanical properties, leading to slight changes to its surface morphology [29]. The addition of nano-ZnO may improve the mechanical strength of films due to the formation of electrostatic bonds between chitosan and ZnO particles that cause the film surface to be rough [30].

### 3.3. Cherry Tomato Preservation Using Chitosan Composite Film

#### 3.3.1. SSC Measurements

SSC mainly include sugars, acids, soluble proteins, vitamins, minerals, and pigments in fruits and vegetables. Since insoluble polysaccharides are hydrolyzed into monosaccharides, fruit SSC increases during maturity and senescence, such that SSC values remain similar to those of fresh produce indicate a longer shelf life. Compared to day 0, there were significant differences of SSC in each group after 15 days of storage (*p* < 0.05). In this study, SSC declined throughout the storage process (Figure 5A). After 15 days of storage, the SSC of the coated cherry tomatoes was 5.17%, which was higher than that of the control group (5.0%). This indicated that the chitosan composite film effectively inhibited respiration and slowed down carbohydrate hydrolysis into sugars leading to a reduction in SSC in cherry tomatoes, which was similar to the study on nano-SiOx/chitosan complex coating on tomatoes [31]. Sammi and Masud reported that the decrease in SSC of cherry tomatoes was probably due to the inhibition of respiratory intensity and reducing metabolic activity of fruits by the coating [32]. However, Ali et al. reported that guar gum and ginseng extract as a coating for sweet cherry, SSC increased steadily during storage, which was contrary to our results [33]. This process may be related to other factors such as the type of fruit, different growth stages and storage conditions.

#### 3.3.2. Color

Significant changes in the color of cherry tomatoes due to composite film coating are crucial to their sensory quality. Generally, *ΔE* in the range of 0 to 1 indicates a color difference invisible to the naked eye, whereas that ranging from 1 to 3 is slightly noticeable, and the color difference can be observed obviously over 3.5. The ripe color of cherry tomatoes is mainly related to pigments, and the color difference varies greatly from immature to mature. During the storage process, there are many factors such as chlorophyll degradation and enzymatic reactions, which can affect the color of fruits. In this study, the overall color difference was larger in the control than coating group (Figure 5B). After 15 days of storage, we observed *ΔE* values of 6.24 and 8.76 in the coating and control groups, respectively. Compared with other days of storage, the two groups were all significantly different (*p* <0.05). This indicated that the chitosan composite film effectively reduced the color change of cherry tomatoes, which may be caused by the low O_2_ and high CO_2_ concentrations that can reduce the activities of enzymes [34].

#### 3.3.3. Respiratory Intensity

Changes in the respiratory intensity of cherry tomatoes during storage at 20 ± 5 °C and 70–75% relative humidity are shown in Figure 5C. Tomatoes exhibit climacteric respiration, showing a clear respiration peak after growth stops followed by a gradual decrease in respiration intensity with age; our findings were consistent with these trends. The coating group reached a peak of 13.47 mg/(kg h) on day 3, and then decreased to 10.83 mg/(kg h) on day 15. A significant difference (*p* < 0.05) was observed on day 15, which was compared to day 3. The chitosan–nano-ZnO composite film did not significantly delay the appearance of the cherry tomato respiratory peak. However, the coating group had a lower respiratory rate than the control group, and respiratory intensity was higher in the control than coating group on day 15, at about 1.34 mg/(kg h). This indicated that the coating formed a barrier on the fruit surface, inhibiting respiratory intensity to some extent and slowing the aging process. In addition, cherry tomatoes coated with chitosan–nano-ZnO composite film showed a lower respiratory rate and higher SSC than those coated in pure chitosan film. Chitosan-nano-ZnO composite film can reduce the level of respiration and slow down the metabolic activity and maturation process of cherry tomatoes. Das et al. reported that coating of fruit by film can reduce respiration, slow synthesis and metabolite utilization as well as carbohydrate hydrolysis into sugars, thereby reducing SSC [35]. Wu et al. reported that the film coating fruits could inhibit gas exchange, thus reducing O_2_ concentration, slowing down V_C_ oxidation and maintaining a high content of V_C_ in coated fruits [36]. Dong et al. reported that the respiration rate of sweet cherry coated with guar gum and ginseng extract was lower than those with other treatments [37]. Similarly, a previous study reported the formation of a nano-SiOx composite film on the surface of Chinese cherry, which limited gas flow, thereby inhibiting respiratory intensity [38]. Thus, in the present study, the composite film slowed nutrient consumption, exerting a preservative effect.

#### 3.3.4. Ascorbic Acid Content

Cherry tomatoes are rich in vitamin C, which must be retained to the greatest extent during storage. In this study, ascorbic acid content generally increased and then decreased throughout the storage period, but varied widely among individual cherry tomatoes (Figure 5D). After 15 days of storage, the ascorbic acid content of the coating group was 609.19 μg/g, which was lower than that of the control group. The coating may have inhibited ascorbic acid synthesis, thereby slowing changes in ascorbic acid content. At the same time, the chitosan–nano-ZnO composite film may block the gas exchange with the external environment, resulting in the decrease in ascorbic acid content. Petriccione et al. reported that sweet cherries coated with 1.5% chitosan were lower in ascorbic acid content than the control during 14 days of storage, which was similar to our result [39].

### 3.4. Inoculation Antimicrobial Assay

The effects of the pure and composite chitosan preservative films on the concentrations of *A. acidoterrestris*, *S. aureus*, *E. coli*, and *Salmonella* in cherry tomato samples are shown in Figure 6. The composite films (0.6% nano-ZnO) exhibited an antibacterial effect. After 15 days of storage, each of the four bacteria was less abundant in the coating than control group. The counts of *A. acidoterrestris*, *S. aureus*, *E. coli*, and *Salmonella* were 8.38, 7.06, 8.99, and 8.75 log CFU/mL in the control group, and 6.62, 5.80, 6.97, and 5.76 log CFU/mL in the coating group, respectively. Additionally, compared to the storage of day 0, the amount of *A. acidoterrestris*, *E. coli* and *Salmonella* have significant differences (*p* < 0.05) on day 15. Laila et al. suggested the coatings showed good antimicrobial activity in packed okra samples during the twelve days of storage [4]. Thus, the composite film showed a good inhibitory effect against *A. acidoterrestris*, *S. aureus*, *E. coli* and *Salmonella* during cherry tomato storage. Sou et al. reported that the count of *S. aureus* in meat with CMC-Na films increased by 5.43 log CFU g^−1^, while that of *S. aureus* with ZnO nanoparticle-coated CMC-Na films only increased by 2.27 log CFU g^−1^ after 14 days [40]. Jin et al. reported that the count of *Listeria monocytogenes* and *Salmonella* in liquid egg white with 1.12 mg ZnO/mL decreased from 3.9 to 1.4 and 4.5 to 3.5 log CFU/mL after 8 days, respectively [41]. One study showed that the antibacterial activity of nano-ZnO against *S. aureus* was better than that against *E. coli*; nonetheless, nano-ZnO greatly improved the antibacterial properties of chitosan film [42]. Furthermore, the reactive oxygen species (ROS) generation was proposed to be a widely accepted antibacterial mechanism of nano-ZnO. ROS, such as O_2_^-^, -OH and H_2_O_2_, are usually produced by nano-ZnO under light radiation [43]. Among them, H_2_O_2_ is generally regarded as the main harmful factor. H_2_O_2_ can destroy the cytoderm and release the DNA and protein in the cytomembrane, so as to achieve the purpose of sterilization [44].

## 4. Conclusions

In this study, we prepared chitosan composite films with varying nano-ZnO contents and evaluated their mechanical, optical, water solubility and antibacterial properties. We showed that chitosan and nano-ZnO were compatible; the addition of nano-ZnO can improve the TS, EAB, and antibacterial activity of chitosan film. The composite film containing 0.6% nano-ZnO had the strongest antibacterial activity against *A. acidoterrestris*, *S. aureus*, *E. coli*, and *Salmonella.* Considering the experiment results, the optimal concentration of the nano-ZnO was 0.6%. The characteristics of the obtained optimal chitosan composite films were measured by XRD and SEM. These results further proved that the chitosan composite film was successful prepared. Based on this, the obtained composite films were used for the preservation of cherry tomatoes. The results indicated that the chitosan-nano-ZnO composite film coated on the surface of cherry tomatoes can limit gas exchange, slow down the respiratory intensity, maintain the SSC and color, and inhibit the growth of microorganisms on the surface of cherry tomatoes, thereby prolonging the shelf life of tomatoes. We conclude that the chitosan–nano-ZnO composite film represents a natural material that can be applied as a new method for solving storage problems related to cherry tomatoes.

## Figures and Tables

**Figure 1 foods-10-03135-f001:**
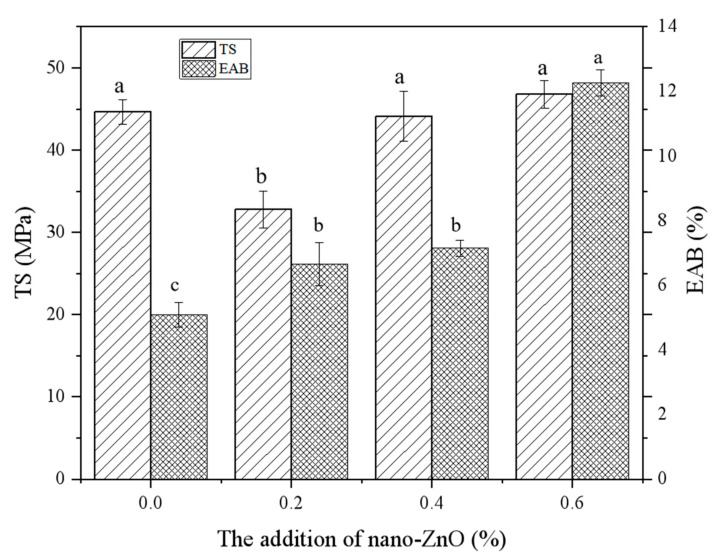
Effect of addition of nano-ZnO on mechanical properties. Different letters in the column indicate significant differences (*p* < 0.05).

**Figure 2 foods-10-03135-f002:**
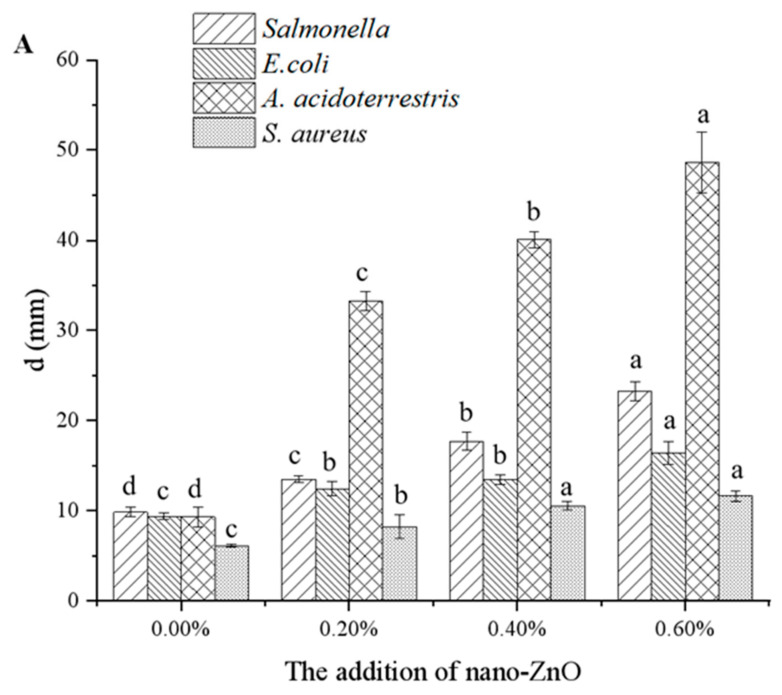
The inhibition zone diameter in different strains (**A**) and the inhibition zone of composite membranes (**B**) under different nano-ZnO contents (0, 0.2%, 0.4%, 0.6%). Lowercase letters in (**A**) indicate the significant differences in different strain (*p* < 0.05). **a**, **b**, **c** and **d** represent the strains of *Salmonella*, *E. coli*, *A. acidoterrestris* and *S. aureus*, respectively, in (**B**).

**Figure 3 foods-10-03135-f003:**
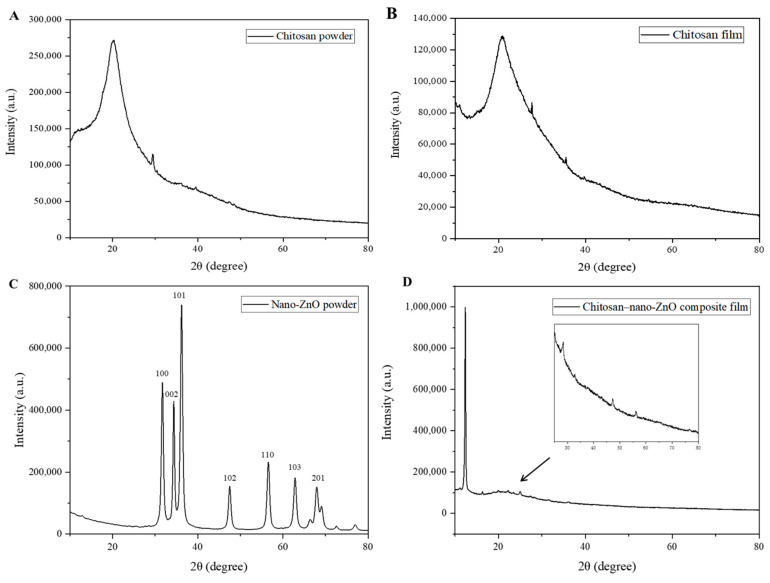
The XRD patterns of chitosan powder (**A**), chitosan film (**B**), nano-ZnO powder (**C**) and chitosan composite film with 0.6% nano-ZnO (**D**).

**Figure 4 foods-10-03135-f004:**
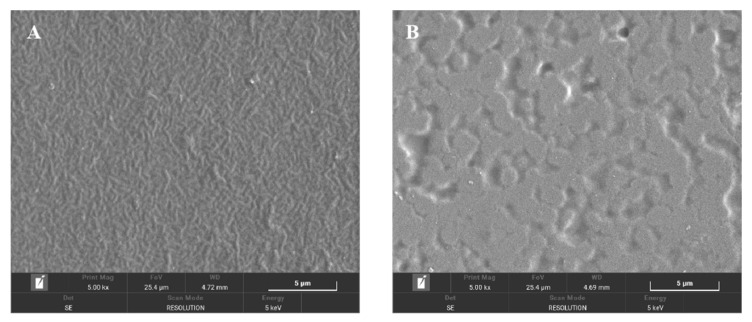
SEM spectrum of chitosan film (**A**) and chitosan composite film with 0.6% nano-ZnO (**B**).

**Figure 5 foods-10-03135-f005:**
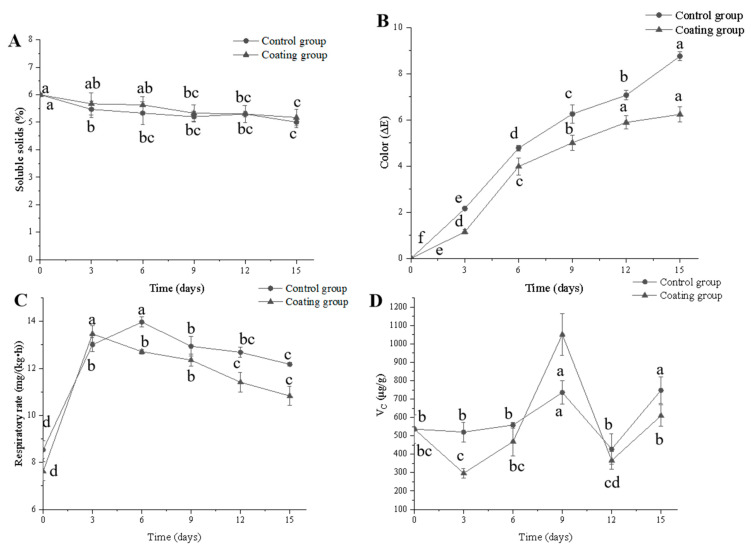
Changes in soluble solids (**A**), color (**B**), respiratory intensity (**C**) and ascorbic acid content (**D**) of cherry tomatoes during storage by different treatments. Different letters indicate significant differences (*p* < 0.05).

**Figure 6 foods-10-03135-f006:**
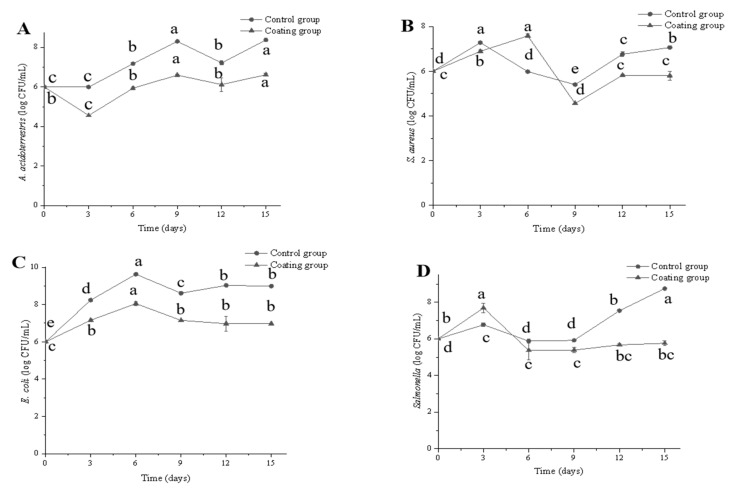
Changes in the concentration of *A. acidoterrestris* (**A**), *S.aureus* (**B**), *E.coli* (**C**) and *Salmonella* (**D**) of cherry tomatoes during storage by different treatments. Different letters indicate significant differences (*p* < 0.05).

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
