# Peer review of "Preparation and Characterization of Chitosan–Nano-ZnO Composite Films for Preservation of Cherry Tomatoes"

_foods, 2021, doi:10.3390/foods10123135_

Round 1
Reviewer 1 Report
Dear authors,
Research on the development of biodegradable packaging to increase the shelf life of products has been significantly improved.
Consider the following suggestions:
Fig 4- standardize the scale in y axis
line 455 - the authors mentioned “previous studies”, but included just one reference . Also, insert the numeric values of bacterial inhibition
of literature compared
line 458-459 - what do you mean? The authors did not
measure the ZnO films activity against ROS.
Here the authors can explain that possibly the nano ZnO films affected (increased?) the ROS formation
in bacterial matrix (double check this information )
Reviewer 2 Report
Thanks to the authors for their diligent attention to the comments made. For my part, I believe that the article has been substantially improved.
Just a few additional minor comments.
L158-160 The information the results of the analysis which lead to the selection of optimal film are presented in results section 3.1 should be given.
L171 correct to “acceleration image”. The fragment „images were captured at magnifications of 5,000 V” doesn’t make any sense.
L221 This should be named “results and discussion”
Reviewer 3 Report
According to the reviewers' suggestions, the article has been improved.
Author Response
Thank you very much for your support to our work. We have revised the article again according to the comments of editors and reviewers.
This manuscript is a resubmission of an earlier submission. The following is a list of the peer review reports and author responses from that submission.
Round 1
Reviewer 1 Report
In my opinion, the reviewed manuscript is not a finished work, it is just a draft of the scientific paper.
A lot of results are missing. In fact, from prepared various ZnO-enriched films (0.2-0.6%) for all of the determined parameters, except mechanical and antimicrobial assays, only results for 0.6% (am I right? This information is also missing) are presented.
What was the number of cherry tomatoes used for each film ?
Numerous references are not related to the discussed statements ([1] is inappropriate, [9] is not about ZnO) or analytical methods ([22] does not give any information about measurements of color difference).
the novelty aspect of the study must be more stressed out. Antimicrobial and mechanical properties of the films can be considered as belonging the M&M section, characterizing the general properties of applied films, without direct application on cherry tomatoes.
Figures captions should be placed below figures. In addition, they are too insufficiently described.
Selected specific comments:
L 44 some information regarding chitosan is needed (origin, material type, etc.)
L79 Brain Heart Infusion (capital letters)
L80 a proper scientific name is Lysogeny broth (LB) , Luria–Bertani is a colloquial name
L93 What amounts specifically?
L104 600 nm is a VIS spectrum, not UV
L129 reference needed
L133 not clear
L198, L326 and others: was it statistically analyzed ?
References are not formatted in accordance with journal requirement.
Author Response
Point 1: A lot of results are missing. In fact, from prepared various ZnO-enriched films (0.2-0.6%) for all of the determined parameters, except mechanical and antimicrobial assays, only results for 0.6% (am I right? This information is also missing) are presented.
Response: Special thanks for your comments. The aim of this study was to prepare chitosan–nano-ZnO composite films with excellent properties and to use it in the preservation of cherry tomatoes, and to reveal its antibacterial mechanism in the process. In order to do that, a series of chitosan–nano-ZnO composite films containing 0.2%, 0.4%, and 0.6% nano-ZnO were prepared and their mechanical, optical, and antibacterial properties were evaluated. Based on the best performance of mechanical properties and antibacterial properties, the mechanical properties and antibacterial properties of the composite film containing 0.6% nano-ZnO was used for the subsequent experiments. The characteristics of composite film containing 0.6% nano-ZnO were further evaluated and their application on cherry tomatoes was also displayed. By this way, we expect to provide new techniques and methods for fruit and vegetable preservation.
Point 2: What was the number of cherry tomatoes used for each film?
Response: In this study, each group containing 9 cherry tomatoes was used for the evaluation of cherry tomatoes quality parameters. Besides, each group containing 9 cherry tomatoes was immersed in inoculation solution for antimicrobial assay. According to your suggestion, we have supplemented this information in the text. (Line 144-145, Line165-166)
Line 144-145: Each group containing nine cherry tomatoes was used for the evaluation of quality parameters.
Line165-166: Each group contains nine cherry tomatoes.
Point 3: Numerous references are not related to the discussed statements ([1] is inappropriate, [9] is not about ZnO) or analytical methods ([22] does not give any information about measurements of color difference).
Response: We are very sorry for our carelessness. Since we have changed the format of the article in the manuscript, the serial number of the references was confused. We have checked all the reference and made modifications.
Point 4: the novelty aspect of the study must be more stressed out. Antimicrobial and mechanical properties of the films can be considered as belonging the M&M section, characterizing the general properties of applied films, without direct application on cherry tomatoes.
Response: Special thanks for your comments. This study focused on the preparation of chitosan–nano-ZnO composite films with good performance that can be used for the preservation of cherry tomatoes. In order to do that, composite films with different nano-ZnO content (0, 0.2%, 0.4%, 0.6%) were obtained and their mechanical properties and antibacterial properties were evaluated. According to the results, composite film with 0.6% nano-ZnO was selected as the best one and used for the preservation of cherry tomatoes. The effect of composite film on the quality of cherry tomatoes was further evaluated. At the same time, the bactericidal effect in practical application was also explored. The result indicated that the prepared chitosan nano-ZnO composite film showed a good preservation effect on cherry tomatoes.
Point 5: Figures captions should be placed below figures. In addition, they are too insufficiently described.
Response: Special thanks for your comments. We have placed the figures captions below figures and added the title of figures. In addition, we have supplemented the statistical analysis of data in the text.
Point 6: L 44 some information regarding chitosan is needed (origin, material type, etc.)
Response: Thank you very much for your comments. According to your suggestion, we have supplemented some information of chitosan in the manuscript.
“Chitosan is the deacetylation of chitin. As an important natural polysaccharide, chitosan has been widely used in biomaterials, healthcare, environment and other industrial fields due to its good biocompatibility, antibacterial activity, and safety [5]. At present, chitosan and its modified materials have been widely applied in food preservation and packaging. For example, chitosan-coated (3% and 4%) smoked herring samples displayed more than 4 log10 CFU/g reduction in aerobic plate count, along with complete suppression of yeast, mold and Enterobacteriaceae count [6]. Meanwhile, chitosan can maintain the phenolic and flavonoid compounds along with the physio-chemical quality of smoked fish during frozen storage. A novel procyanidins-loaded chitosan-graft-polyvinyl alcohol film was prepared and applied for sustaining the antibacterial activity for food packaging [7]. The obtained film exhibited a desirable bio-film inhibition and antibacterial activity against spoilage bacteria and foodborne pathogenic microbes. For the salmon preservation, the film can prevent microorganism contamination and texture deterioration in 10 days. At the same time, essential oils loaded-chitosan nanocapsules have been used for the preparation of thermoplastic starch films [8]. The obtained films can inhibit the growth of Escherichia coli or Bacillus subtillis, and extend the strawberries' shelf life without fungi contamination.” (Line 44-60).
Point 7: L79 Brain Heart Infusion (capital letters)
Response7: We are very sorry for our carelessness. According to your comments, the statement of “Brain heart infusion” has been revised as “Brain Heart Infusion”. (Line 96)
Point 8: L80 a proper scientific name is Lysogeny broth (LB), Luria–Bertani is a colloquial name
Response: We are very sorry for our carelessness. According to your comments, the statement of “Luria–Bertani” has been revised as “Lysogeny broth”. (Line 97)
Point 9: L93 What amounts specifically?
Response: We are very sorry for our inaccurate description. The amount of nano-ZnO is 0.2%, 0.4%, 0.6% of the volume of the solution respectively. According to your comments, the statement has been revised as “different amounts of nano-ZnO (0.2%, 0.4%, 0.6% (w/v)) were added”. (Line 110)
Point 10: L104 600 nm is a VIS spectrum, not UV
Response: We are very sorry for our carelessness. According to your comments, we have revised as “UV–vis spectrophotometer”. (Line 121)
Point 11: L198, L326 and others: was it statistically analyzed?
Response: Special thanks for your comments. The data of experiments has been statistically analyzed. And we have added statistical analysis in the text. (Line 190-198, 213-214, 295-298, 321-323, 344-345, 354-355, 396-398)
Line 190-198: Compared with the pure chitosan film, the TS values of composite films containing 0.2% and 0.4% nano-ZnO decreased by 26.57% and 1.21%, while that of the composite films containing 0.6% nano-ZnO increased by 4.61%, respectively. The TS values of composite films containing 0.4% and 0.6% nano-ZnO were no significant difference (p > 0.05). The EAB values of composite films containing 0.6% nano-ZnO were significantly higher (p < 0.05) than that of the pure chitosan film and composite films containing 0.2% and 0.4% nano-ZnO. The EAB values of composite films containing 0.2%, 0.4%, and 0.6% nano-ZnO increased by 30.84%, 40.39%, and 140.86%, respectively.
Line 213-214: There were obvious differences between each group of composite films and pure chitosan film (p < 0.05).
Line 295-298: The antimicrobial activity of composite films containing 0.6% nano-ZnO showed appreciable difference (p < 0.05) towards four strains when compared to the pure chitosan film (Figure 5A)
Line 321-323: Compared to 0 day, there were significant differences of SSC in each group after 15 days of storage (p < 0.05).
Line 344-345: Compared with other days of storage, the two groups were all significantly different (p <0.05).
Line 354-355: A significant difference (p <0.05) can be observed on day15, which was compared to the day3.
Line 396-398: And compared to the storage of day 0, the amount of A. acidoterrestris, E. coli and Salmonella have significant differences (p <0.05) on day 15.
Point 12: L129 reference needed
Response 12: Special thanks for your suggestion. According to your comments, we have added reference in the text.
Point 13: L133 not clear
Response 13: We are very sorry for our inaccurate description. We have added this information of “Cherry tomatoes were crushed in a blender, homogenized and then measured by saccharometer” in the text. (Line 158-159)
Point 14: References are not formatted in accordance with journal requirement.
Response: Thank you very much for your good comments. We have corrected the format of the journal.

Reviewer 2 Report
The novelty of the paper is low. Previous reports show that the addition of nano-ZnO to chitosan enhance the mechanical properties of the films. On the other hand, there are some reports that evaluate the antimicrobial properties of nano-ZnO composite film. The application on cherry tomato is interesting.
Some aspects are not clear enough and some explanations are necessary.
Line 19-20: Alicyclobacillus acidoterrestris, Staphylococcus aureus, Escherichia coli, and Salmonella in italics
Line 105 Please add the explanation of XRD
Line 107 Please add the explanation of SEM
Line 124 each sample was immersed in chitosan solution ¿why? ¿How cherry tomatoes were packaged in the films? In line 95-96 the authors described: “Finally, 20 mL of each prepared solution was added to a 10 cm × 10 cm culture dish and air dried at room temperature to obtain uniform films”. The objective of the paper is to evaluate the mechanical properties of the films. The films generated are not used in cherry tomato packaging. The cherry tomato was immersed in chitosan solution. This point needs to be clarified too in the manuscript.
Line 151 the authors described Student’s t-test to evaluate significant differences between treatments were used. However, no statistical analyses in the data. Please indicate statistical differences in Figures.
All data in the figures 1, 5, 6 and 7 were represented with average and standard deviation or standard error? (This point needs to be clarified too in the manuscript).
Captions are mandatory for figures and are added below figures.
Line 318: Add explanation of a,b,c,d in Figure 5B
Line 373 Please confirm the content of Reference 47
Line 471: References should be correct according to the format of the journal.
Author Response
Point 1: Alicyclobacillus acidoterrestris, Staphylococcus aureus, Escherichia coli, and Salmonella in italics
Response 1: We are very sorry for our carelessness. According to your suggestion, “Alicyclobacillus acidoterrestris, Staphylococcus aureus, Escherichia coli, and Salmonella” has been revised as “Alicyclobacillus acidoterrestris, Staphylococcus aureus, Escherichia coli, and Salmonella”. (Line 19-20)
Point 2: Line 105 Please add the explanation of XRD
Response 2: Special thanks for your suggestion. According to your suggestion, we have added the explanation of XRD as “The samples were analyzed between 10 and 80° with increments of 2°/min” in the text. (Line 124)
Point 3: Line 107 Please add the explanation of SEM
Response 3: Thank you very much for your comments. According to your suggestion, we have added the explanation of SEM as “Gold sputtering was performed by an MSP-IS ion sputtering instrument. Using an accelerating voltage of 5 kV, images were captured at magnifications of 5,000V” in the text. (Line 126-128)
Point 4: Line 124 each sample was immersed in chitosan solution why? How cherry tomatoes were packaged in the films? In line 95-96 the authors described: “Finally, 20 mL of each prepared solution was added to a 10 cm × 10 cm culture dish and air dried at room temperature to obtain uniform films”. The objective of the paper is to evaluate the mechanical properties of the films. The films generated are not used in cherry tomato packaging. The cherry tomato was immersed in chitosan solution. This point needs to be clarified too in the manuscript.
Response 4: We apologize for the unclear description of the details. We have revised sentences as “Then, the cherry tomato samples of the coating group were immersed in chitosan solution with 0.6% nano-ZnO, shaken gently for 1 min, and air-dried in a biological safety cabinet for 1 h. The control groups were immersed in distilled water, and the other steps remain the same as the coating group.” (Line 146-149). Besides, according to your comments, we have added “The characterization and antimicrobial properties of the prepared films were further evaluated” in the text.” (Line 113-114)
Point 5: Line 151 the authors described Student’s t-test to evaluate significant differences between treatments were used. However, no statistical analyses in the data. Please indicate statistical differences in Figures.
Response 5: We apologize for our carelessness. According to your suggestion, we have revised description as “All statistical analysis was performed using SPSS 18.0 (SPSS Inc., USA). The obtained data were reported as the mean ± standard deviation (n≥3). We performed one-way analysis of variance (ANOVA) followed with Duncan's multiple test to evaluate significant differences between treatments (p<0.05). All plots were created using OriginPro 8.0 software (OriginLab, USA)” in the text (Line 177-181). In addition, we have added and revised statistical analyses in the paper (Line 190-198; Line 295-297; Figure 1).
Line 190-198: Compared with the pure chitosan film, the TS values of composite films containing 0.2% and 0.4% nano-ZnO decreased by 26.57% and 1.21%, while that of the composite films containing 0.6% nano-ZnO increased by 4.61%, respectively. The TS values of composite films containing 0.4% and 0.6% nano-ZnO were no significant difference (p > 0.05). The EAB values of composite films containing 0.6% nano-ZnO were significantly higher (p < 0.05) than that of the pure chitosan film and composite films containing 0.2% and 0.4% nano-ZnO. The EAB values of composite films containing 0.2%, 0.4%, and 0.6% nano-ZnO increased by 30.84%, 40.39%, and 140.86%, respectively.
Line 295-297: The antimicrobial activity of composite films containing 0.6% nano-ZnO showed appreciable difference (p < 0.05) towards four strains when compared to the pure chitosan film.
Point 6: All data in the figures 1, 5, 6 and 7 were represented with average and standard deviation or standard error? (This point needs to be clarified too in the manuscript).
Response 6: Thank you very much for your good comments. Our data in the figures were represented with average and standard deviation. According to your suggestion, we have supplemented “The obtained data were reported as the mean ± standard deviation (n≥3)” in the text (Line 177-178).
Point 7: Captions are mandatory for figures and are added below figures.
Response 7: Special thanks for your comments. We have added captions in the text.
Point 8: Line 318: Add explanation of a,b,c,d in Figure 5B
Response 8: Thank you very much for your good comments. According to your comments, we have added explanation of a, b, c, d in Figure 5B and in the text. (Line 283-290)
Line 283-290: The inhibition zone diameters and disc diffusion study in different strains was showed in Figure 5A and B, where a, b, c and d represented the strains of Salmonella, E. coli, A. acidoterrestris and S. aureus, respectively. The inhibition zones of pure chitosan film against Salmonella, E. coli, S. aureus, and A. acidoterrestris measured 9.85±0.50, 9.37±0.36, 9.30±1.13, and 6.13±0.20 mm, respectively, perhaps due to the antimicrobial properties of chitosan. The disc diffusion study showed that all films had significant antibacterial effects on the selected microbial strains (Figure 5B).
Point 9: Line 373 Please confirm the content of Reference 47
Response 9: We are very sorry for our carelessness. We have revised the sentence as “Similarly, a previous study reported the formation of a nano-SiOx composite film on the surface of Chinese cherry, which limited gas flow, thereby inhibiting respiratory intensity [42].” (Line 370-372) and corrected Reference 47.
Point 10: Line 471: References should be correct according to the format of the journal
Response 10: Thank you very much for your good comments. We have corrected the the format of the journal
Reviewer 3 Report
Just a few comments for improvement:
L 307-308 - Any idea why your results contrasted the publication cited? Include the possibilities.
Figure 1: double check the statistical difference within the EAB ones - I can see visible differences.
Figures: 1 - Do not insert the caption in the Figure. Include figures after the figure captions. Figure captions are included after the references section. 2-In a figure with multiple graphics, include the whole Figure - all graphics - in a single page)
Conclusion: that would be good the authors include the optimal concentration of nano-Zn used against microorganisms
Author Response
Point 1: L 307-308 - Any idea why your results contrasted the publication cited? Include the possibilities.
Response 1: Thank you very much for your comments. According to your suggestion, we have added description “It may be caused by the presence of a thick layer of peptide glycans in the cell wall of S. aureus” in the text. (Line 304-305)
Point 2: Figure 1: double check the statistical difference within the EAB ones - I can see visible differences.
Response 2: We are very sorry for our carelessness. We have revised Figure 1 in the text.
Point 3: Figures: 1 - Do not insert the caption in the Figure. Include figures after the figure captions. Figure captions are included after the references section. 2-In a figure with multiple graphics, include the whole Figure - all graphics - in a single page)
Response 3: Thank you very much for your good comments. According to your suggestion, we have revised figures in the text.
Point 4: Conclusion: that would be good the authors include the optimal concentration of nano-ZnO used against microorganisms
Response 4: Thank you very much for your good comments. According to your suggestion, we have added “Considering the experiment results, the optimal concentration of the nano-ZnO was 0.6%”. (Line 413-414)

Round 2
Reviewer 1 Report
After carefully reading the revised manuscript, and despite the changes made, initial main concerns remain. From my point of view, the experimental design is weak regarding the functional evaluation of films which does not allow to draw the clear conclusions regarding this aspect.
The whole experimental protocol, and subsequently the whole manuscript, should be divided into two separate trails. The first one should be focused on the examination of the general physical properties of different films and the second one (trail two) should show only the results for subsequent analyses performed on the selected film only.
This raised the main question regarding the presented results. If the selection criteria was the mechanical properties only, why the next experimental assay was the evaluation of the optical properties of all films ? Finally, what is the purpose for measurement of their antimicrobial properties? It doesn’t make any sense. All these additional analyzes had absolutely no influence on the selection of the optimal film. Also for analyses performed in the second part of the manuscript, as measurements are performed on the same samples in different time points – repeated measurements ANOVA should be performed instead of simple ANOVA followed with Duncan test.
All of the above remarks clearly show that the whole experimental schedule was incorrectly designed and, therefore, the whole experiment was improperly conducted.
Reviewer 2 Report
The manuscript has been sufficiently improved to warrant publication in Foods.